# MicroRNA-155 Modulates Macrophages’ Response to Non-Tuberculous Mycobacteria through COX-2/PGE2 Signaling

**DOI:** 10.3390/pathogens10080920

**Published:** 2021-07-21

**Authors:** Zhihong Yuan, Zohra Prasla, Frances Eun-Hyung Lee, Brahmchetna Bedi, Roy L. Sutliff, Ruxana T. Sadikot

**Affiliations:** 1VA Nebraska Western Iowa Health Care System, Omaha, NE 68105, USA; 2Division of Pulmonary, Critical Care & Sleep, Department of Internal Medicine, University of Nebraska Medical Center, Omaha, NE 68198, USA; 3Division of Pulmonary, Allergy, Critical Care, and Sleep Medicine, Department of Medicine, Emory University, Atlanta, GA 30322, USA; zprasla@mednet.ucla.edu (Z.P.); f.e.lee@emory.edu (F.E.-H.L.); bbedi@emory.edu (B.B.); rsutlif@emory.edu (R.L.S.)

**Keywords:** non-tuberculous mycobacteria, *Mycobacterium avium*, COX-2, PGE2, miR-155

## Abstract

Non-tuberculous mycobacteria (NTM) have been recognized as a causative agent of various human diseases, including severe infections in immunocompromised patients, such as people living with HIV. The most common species identified is the *Mycobacterium avium*-intracellulare complex (MAI/MAC), accounting for a majority of infections. Despite abundant information detailing the clinical significance of NTM, little is known about host–pathogen interactions in NTM infection. MicroRNAs (miRs) serve as important post-transcriptional regulators of gene expression. Using a microarray profile, we found that the expression of miR-155 and cyclo-oxygenase 2 (COX-2) is significantly increased in bone-marrow-derived macrophages from mice and human monocyte-derived macrophages from healthy volunteers that are infected with NTM. Antagomir against miR-155 effectively suppressed expression of COX-2 and reduced Prostaglandin E**_2_**(PGE2) secretion, suggesting that COX-2/PGE2 expression is dependent on miR-155. Mechanistically, we found that inhibition of NF-κB activity significantly reduced miR-155/COX-2 expression in infected macrophages. Most importantly, blockade of COX-2, E-prostanoid receptors (EP2 and EP4) enhanced killing of MAI in macrophages. These findings provide novel mechanistic insights into the role of miR-155/COX-2/PGE2 signalling and suggest that induction of these pathways enhances survival of mycobacteria in macrophages. Defining host–pathogen interactions can lead to novel immunomodulatory therapies for NTM infections which are difficult to treat.

## 1. Introduction

Non-tuberculous mycobacteria (NTM) represent over 190 species and subspecies, and have been recognized as a causative agent of various human diseases, affecting both pulmonary and extrapulmonary sites. As ubiquitous environmental organisms, NTM can cause chronic pulmonary infection, particularly in individuals with pre-existing inflammatory lung disease, such as cystic fibrosis, bronchiectasis, and chronic obstructive pulmonary diseases (COPD) [1,2,3,4,5]. The number of NTM infections is rising world-wide, and in the U.S. the number of NTM infections currently exceeds that of tuberculosis. It is estimated that most common pathogens, such as Mycobacterium kansasii, and Mycobacterium xenopi, *Mycobacterium avium* complex (MAC, including *M. avium, M. intracellulare* and *M. chimaera*), and *M. abscessus*), account for the majority of disseminated infections, including 90% of the total number of reported cases of NTM-pulmonary disease (NTM-PD) [6,7]. Despite the abundance of information detailing the clinical significance of NTM, management of non-tuberculous mycobacterial pulmonary disease, including its epidemiology, diagnosis, treatment, and prevention remain to be characterized [8]. With increased resistance in NTM infections, new treatment options are urgently needed to treat these refractory infections. Relatively little is known about innate immunity to NTM infection, and host–pathogen interactions are poorly defined [9,10].

Cyclo-oxygenase (COX) is an enzyme that catalyzes the synthesis of prostaglandins (PGs) from arachidonic acid. COX-2 is induced upon stimulation by inflammatory stimuli such as LPS, and plays a key role in inflammation. Previous studies have investigated the role of COX-2, and distal prostaglandins, PGE2 and PGD2 in the pathogenesis of bacterial, fungal, and viral infections [11,12,13,14,15,16]. Mycobacterium tuberculosis (MTB) can result in increased expression of cyclooxygenase-2 (COX-2) with production of PGE2, which can modulate the innate immune capacity of the host [17,18,19]. COX-2 has shown to alter the antimycobacterial immune responses in Mtb via multiple mechanisms, such as cooperating with IL-1 to limit production of type I interferon [20]. MPT83, a secreted mycobacterial lipoprotein, induces apoptosis of macrophages by activation of the TLR2/p38/COX-2 axis in macrophages [21]. Furthermore, induction of COX-2 alters the repair of plasma membranes by promoting apoptosis of host macrophages [22,23]. COX-2 induction also modulates macrophage responses to MTB through autophagy [24]. The effects of COX-2 are mediated by production of prostanoids PGE2 and PGD2. In general, PGE2 is immunosuppressive [11,12,15,16], whereas PGD2 has shown to have an immunomodulatory response in some infections [16,25]. There is little to no information about whether infections with NTM alter host responses through COX-2 and the mechanisms by which COX-2 is induced.

MiRNAs are 18-22-nucleotide noncoding RNA molecules that regulate gene expression post-transcriptionally through base-pairing with mRNAs, resulting in either translational repression or mRNA degradation, and thus regulate the expression of multiple genes at a post-transcriptional level [26,27]. Besides inhibitory effects on mRNA stability and translation, miRNAs can also activate gene expression. Several miRNAs have shown to play a key regulatory factor in the innate immunity to infections, including MTB [28,29,30,31,32]. Information about the role of miRNA in NTM infections is scant [33]. MiR-155 is a pleiotropic microRNA that plays a crucial role in modulation of the inflammatory response [34,35,36]. MiR-155 expression is increased in MTB-infected macrophages and lungs of mice with MTB infection [37]. Studies have showed that miR-155 and cyclooxygenase (COX)-2 are both elevated in patients with TB, colorectal cancer, and asthma [38,39,40]. Interestingly, miR-155 enhances COX-2 expression and is an established regulator of inflammation [38]. However, whether induction of miR-155 and COX-2 can play a role in immune response to NTM infections is not known.

In this study, we investigated the expression of miRNA profiles in MAI-infected murine and human macrophages. Our data show that selected miRNAs are upregulated in macrophages in response to MAI infection. We found that expression levels of miR-155 and COX-2 are significantly increased in MAI-infected macrophages. Given the importance of miR-155 and COX-2 in the innate immune response, we further investigated whether expression of COX-2 is dependent on the upregulation of miR-155. For the first time, our data show that infection of macrophages with MAI upregulates miR-155 in an NF-kB-dependent manner and is responsible for expression of COX-2 and production of PGE2. Most importantly, inhibition of COX-2 and the blockade of EP2 and EP4 decreased the survival of NTM in macrophages.

## 2. Results

### 2.1. MiR-155 Is Upregulated in Bone-Marrow-Derived Macrophages (BMDM) Infected with MAI

To delineate whether the NTM-induced inflammatory response was regulated by miRNAs, we infected bone-marrow-derived macrophages (BMDM) from wild-type C57 BL/6J mice with MAI or vehicle. Total RNA was extracted, and miRNA expression was determined by using murine inflammatory response array-based miScript.

As shown in Figure 1A,B, microarray data from mouse bone-marrow-derived macrophages infected with MAI (MOI:25) showed an altered expression profile of miRNAs. We found that expression of a couple of miRNAs significantly upregulated, including miR-155 (fold change: 4.39), miR-98-5p (fold change:3.69), miR-669h-3p (fold change:2.96), miR-9-5p (fold change: 2.87), miR-93-5p (fold change: 2.7), miR-712-5p (fold change: 2.35), miR-669k-3p (fold change: 2.15), let-7f-5p (fold change:2.09), and miR-26-5p (fold change: 2.03), whereas there were decreased expression levels of some miRNAs, including miR-181c-5p (fold change −4.08), miR-126a-5p (fold change −12.01), and miR-19b (fold change −5.39). Next, we performed additional experiments and validated the expression of these miRNAs by performing RT-qPCR and confirmed that they are indeed upregulated. Previously, we and others have shown that miR-155 regulates multiple inflammatory genes through modulation of SOCS1 [35]; hence, we focussed our further studies on miR-155. Expression of miR-155 significantly increased in mouse BMDM as early as 6 h after treatment with MAI at different MOI of 25 and 50, and the elevating trend continued for at least 24 h (Figure 1C). To determine whether this was relevant to human infections, similar experiments were performed in human monocyte-derived macrophages (MDM) from healthy volunteers. Like murine BMDM, we found that the expression of miR-155 was significantly increased in HMDM that were infected with MAI (Figure 1D). Together, these data suggest that MAI induces expression of miR-155 in host macrophages. We hypothesize that this increased induction can modulate the host immune response.

### 2.2. Pro-Inflammatory Mediators Regulated by miR-155 Are Induced in Macrophages Infected with MAI

Next, to delineate the impact of miR-155, we performed a mini micro-array to specifically determine the expression of select genes that are regulated by miR-155. Murine BMDM were infected with MAI at MOI of 25 for six hours, then proinflammatory gene profiles were detected through a profile PCR array. Scatter plot showed that MAI induced the expression of proinflammatory genes, including COX-2 and NF-κB, as well as robust expression of proinflammatory cytokines, including TNF, IL-1β, and IL-6, as shown in Figure 2A. Corresponding data demonstrated significant increased expression of proinflammatory genes, including IL-1β (fold change: 245.28), TNF (fold change: 125.4), IL-6 (fold change: 18.25), NF-κB (fold change: 2.11), and COX-2 (fold change: 62.88). The expression of these genes was further validated by using RT-qPCR. As suggested by the micro-array, data on expression levels of these pro-inflammatory genes confirmed a significant increase in infected BMDMs (Figure 2B–D). Similar experiments were performed in HMDM infected with MAI (MOI of 25). Like murine BMDM, the expression of pro-inflammatory genes was increased in human MDM infected with MAI (Figure 2E,F). We have previously shown that COX-2 expression is significantly increased in Gram-negative bacterial infections with increased production of PGE_2_, which has immunosuppressive effects [14]. Furthermore, we confirmed that the expression of COX-2 mRNA was increased by infection of macrophages with MAI (Figure 2G). Western blot assays also indicated that expression of COX-2 protein increased upon MAI infection, as shown in Figure 2H. COX-2 expression level in MAI-infected human MDM also demonstrated a similar pattern, as shown in Figure 2I. Collectively, these data show that murine and human macrophages infected with NTM induce expression of pro-inflammatory mediators with increased expression of COX-2.

### 2.3. Induction of COX-2 and Production of PGE2 by MAI Is Dependent on miR-155

We next wanted to investigate whether miR-155 induces the expression of COX-2 in macrophages infected with MAI. MiR-155 plays an essential role in driving the inflammatory response [22]. Inhibition of miR-155 has shown to attenuate lung injury in models of lung inflammation and modulate immune response through regulation of multiple genes. Previous studies have shown that miR-155 can enhance the expression of COX-2. To investigate whether miR-155 contributes to the expression of COX-2, we treated macrophages with miR-155 antagomirs (miR-155i) to silence the expression of miR155. BMDM were transfected with antagomir against miR-155 or control antagomirs. Twenty-four hours post-transfection, cells were infected with MAI (MOI of 25) for 24 h. COX-2 expression was measured by RT-qPCR and Western blot analysis. Cells that were treated with control antagomirs showed a significant induction of COX-2 compared to uninfected cells, whereas cells treated with antagomir against miR-155 showed a significant reduction in the expression level of COX-2 mRNA and protein (Figure 3A,B).

We and others have shown that elevated COX-2 expression in infections has had immunosuppressive effects through production of PGE2 [14,41]. Therefore, we measured the levels of PGE2 from supernatants of infected macrophages by ELISA. As shown in Figure 3C, there was an increased production of PGE2 in supernatants of macrophages that were infected with MAI and treated with control antagomir. Cells treated with antagomir against miR-155 showed a significant reduction in levels of PGE2 (Figure 3C). Together, these data show that induction of COX-2 and production of PGE2 in response to MAI infection is dependent on miR-155.

Premature miRNAs (pre-miRNAs) are cleaved by ribonuclease III and transported into a miRNA-induced silencing complex (miRISC) to function as a mature miRNA. The core of miRISC is formed by the Argonaute (Ago) proteins capable of mediating endonucleolytic cleavage of the target mRNA, which requires complementarity between a miRNA and target site [42]. Because miRNAs activate the RNA-induced silencing complex to suppress complementary mRNAs, to confirm the direct binding of miR-155 to mRNA encoding COX-2, we performed an RNA-binding protein immunoprecipitation (RIP) assay using an anti-Ago2 antibody, which is the key component of the functional RNA-induced silencing complex that cleaves the target strand [43]. Anti-Ago2 RNA immunoprecipitation experiments show a significant enrichment of miR-155 compared to normal IgG immunoprecipitates as the input control, and an enrichment of COX-2 as well, as shown in Figure 3D,E. These results suggest that miR-155 directly regulates COX-2 expression.

### 2.4. MiR-155 and COX-2 Expression Are Transcriptionally Regulated by NF-kB in Macrophages Infected with MAI

Next, we investigated the mechanism by which miR-155 is induced in macrophages in response to MAI infection. MiR-155HG (pri-miR-155) is transcriptionally regulated by NF-κB, AP-1, and ETS-1. NF-κB is a transcription factor that is a master regulator of pro-inflammatory response and expression of cytokines. Previous studies have shown that NF-κB transcriptionally regulates expression of miRNAs [44]. The proximal promoter region of MIR-155HG has several NF-κB binding sites. We therefore questioned whether NF-κB-regulated miR-155 could modulate the COX-2-PGE2 signalling pathway. We questioned whether the expression of miR-155 is regulated by NF-κB in response to MAI infection. BMDM were treated with BMS (10 µM) prior to infection with MAI for 6 and 24 h. Control cells were treated with vehicle and infected with MAI. The expression of miR-155 was detected by RT-qPCR. Cells that were treated with BMS and infected with MAI showed a significantly reduced expression of miR-155 compared to control cells that were infected with MAI (Figure 4A). The expression of COX-2 was also determined by RT-qPCR and Western blot analysis. Like miR-155, the expression of COX-2 was significantly reduced in cells that were infected with MAI and treated with BMS compared to control cells that were infected with MAI (Figure 4B,C). Together, these data show that expression of miR-155 and COX-2 is regulated by NF-κB in response to MAI infection.

### 2.5. Inhibition of COX-2, EP2 and EP4 Enhances Killing of MAI in Macrophages

We and others have shown that COX-2 activation and production of lipid mediators play a pivotal immunomodulatory role in bacterial infections [13,14,15,45]. PGE2 has immunosuppressive effects and impairs bacterial clearance through activation of EP2 and EP4 receptors [11,15,46]. COX-2 and PGE2 signalling can modulate immune response to *M. tuberculosis* [47] infection. To ascertain whether inhibition of COX-2 exerts modulatory effects in MAI-infected macrophages, we used the specific COX-2 inhibitor NS398. Bone-marrow-derived macrophages were treated with NS398 (10 µM) and then infected with MAI (MOI 25). First, we assessed the expression of COX-2 by Western blotting. As shown in Figure 5A, cells that were treated with NS398 showed a significantly reduced expression of COX-2 compared to cells that were treated with vehicle. Next, we assessed the ability of macrophages to kill MAI in response to COX-2 inhibition. As shown in Figure 5B, BMDM that were treated with NS398 showed enhanced killing of MAI compared to control cells treated with vehicle. COX-2 mediates immunosuppressive effects through production of PGE2.

Since our data show that production of PGE2 is significantly increased in macrophages infected with MAI, we wanted to investigate whether the blockade of downstream signalling through PGE2 could also mediate a beneficial effect. PGE2 mediates its action via G protein-coupled receptors, and four subtypes of the EP receptor for PGE2 have been described previously [48]. Previous studies have shown that immunosuppressive effects of PGE2 are mediated through EP2 and EP4 signalling [15,49,50]. Therefore, we investigated the effects of EP2 and EP4 blockade in macrophage responses to MAI. Macrophages were treated with EP2 and EP4 receptor blockers (AH6809, AH 23848) (10 µM) prior to infection with MAI (MOI of 25). As shown in Figure 5C, cells that were treated with EP2 and EP4 blockers showed reduced survival of MAI. Collectively, these data show that inhibition of COX-2 and downstream effectors of PGE2 signalling alter the survival of MAI in macrophages. Whether these beneficial effects are seen in vivo will need further investigation.

## 3. Discussion

NTM are gram-positive, acid-fast, aerobic bacilli that are ubiquitous in the environment. Although NTM infections have surpassed tuberculosis in Western countries, there is a lack of understanding of host–pathogen interactions in NTM infections. The immune responses generated to mycobacterial infections are determined by host–pathogen interactions and involve multiple cell types. Infection with MAI leads to generation of immune-regulatory molecules which play a crucial role in triggering various downstream signalling pathways. Recent studies have highlighted the role of miRNA in host immune responses [51]. Mycobacteria can alter host immune responses to suit their own benefit which include modulating host miRNAs. These miRNAs bind to complementary target mRNAs, which can regulate a complex set of immune interactions [52]. Here, we have used microarray profile to identify miRNAs which have the potential to modulate immune response to MAI. Our results, for the first time, demonstrate that induction of miR-155 by MAI alters the ability of macrophages to clear NTM through COX-2/PGE2 signalling.

Macrophages play an important role and provide the first line of defence against mycobacterial infections. They can engulf and kill mycobacteria by initiating an inflammatory response. Macrophage host responses to mycobacteria, particularly MTB, has been extensively studied; however, the host response to NTM is less well-characterized [53]. Antimicrobial responses of macrophage comprise of activation of multiple signalling pathways, including induction of miRNAs which can regulate key effector molecules. We found that select miRNAs, miR-155, miR-98, miR-669, miR-9, and let-7i are upregulated in response to MAI. Accumulating evidence has indicated that miR-155 plays an important role in the regulation of inflammation and immunity. A previous study has suggested that miRNA-155 in peripheral blood mononuclear cells is upregulated in patients with tuberculosis [54].

The precise role of miR-155 in macrophages during mycobacterial infection has not been fully defined. Studies have demonstrated that miR-155 enhances production of pro-inflammatory molecules, such as TNF-α in human macrophages in response to mycobacteria, which enables them to eliminate intracellular mycobacteria. These effects of miR-155 are mediated through regulation of SOCS1 [35,55]. There are several other mechanisms by which miR-155 can modulate the immune response in mycobacterial infections. MiR-155 inhibits killing of MTB through regulation of autophagy [56]. In another study, Rothchild et al. showed that miR-155 promotes cell survival and propagation of bacteria in macrophages through the SHIP1/protein kinase B (Akt) pathway [37]. In M. bovis (BCG) infection, miR-155 was responsible for the fate of infected macrophages, implicating a role for miR-155 in orchestrating cellular reprogramming during immune responses to mycobacterial infection [5]. Our study is the first to show that miR-155 alters the ability of macrophages to kill MAI through induction of COX-2 and production of PGE2. These data implicate a novel role for miR-155 in mycobacterial survival in macrophages. Antagomirs of miR-155 can be potential therapeutic targets to modulate host responses in mycobacterial infections.

Cyclooxygenase is the rate-limiting enzyme in the biosynthesis of prostaglandins and related eicosanoids from the arachidonic acid metabolism [57]. COX-2 is induced in macrophages and epithelial cells by many of the pro-inflammatory stimuli, including bacteria, viruses, and fungi [13,14,15,58,59]. We and others have shown COX-2 is induced in bacterial infections, such as *P. aeruginosa,* which can modulate host responses [14,15]. However, the mechanisms by which COX-2 is induced by pathogens is not fully characterized. Previous studies have shown that induction of COX-2 is dependent on NF-kB and can occur through MAP kinase signalling. In this study, we showed that miR-155 is a critical mediator for induction of COX-2. Recent studies have shown a relation between induction of miR-155 and pro-inflammatory mediators, including COX-2 [38,39]. In vitro studies in human airway smooth muscle cells have shown that miR-155 enhances COX-2 expression [40]. In a murine model of allergen-induced oxidative stress, simulating asthma, Qui et al. showed that miR-155 regulates the inflammatory response through induction of COX2. More recently, Zhang et al. showed that miR-155 knockdown inhibited the IL-1β, IL-6, and TNF-α and inflammatory enzymes (iNOS and COX-2), and alleviated ischemia-reperfusion injury by targeting MafB [33]. Our data suggest that in MAI, infection leads to induction of COX-2 and is dependent on miR-155. These data have important implications because targeting miR155 for host immunomodulation can provide a novel strategy to enhance host defences against mycobacterial infections.

Lipid mediators and prostanoids have emerged as potent modulators of innate immune responses to a variety of pathogens [16]. Modulation of immune response by COX-2 is largely related to an increased production of PGE2, which is immunosuppressive in animal models of bacterial pneumonias and sepsis [60,61,62]. PGE2 suppresses anti-microbial activity of alveolar macrophages via the EP2 and EP4 receptor. Therefore, we investigated whether the effects of PGE2 are mediated via these receptors in MAI-infected macrophages. We found that inhibition of COX-2, EP2, and EP4 in macrophages enhances killing of MAI. These data suggest that inhibition of COX-2, EP2, and EP4 receptors can be used as immunomodulators for treatment of MAI infections. However, further studies are needed to establish the role of these signalling pathways in MAI infections in vivo.

In summary, this study shows that infection of macrophages induce expression of select miRNAs and pro-inflammatory mediators. Our data are the first to show that induction of miR-155 in response to MAI is dependent on NF-kB and increases the expression of COX-2 with production of PGE2. The ability of macrophages to clear MAI is inhibited by induction of COX-2 through PGE2 via EP2 and EP4 receptors. Most importantly, induction of miR155 and COX2 during MAI infection contributes to the ability of mycobacteria to survive in macrophages (Figure 6). Thus, inhibition of miR-155, COX-2, EP2, and EP4 receptors can be developed as an adjuvant therapy for mycobacterial infections to enhance the ability of host macrophages to clear NTM. Ongoing animal studies will provide additional data to define the role of these signalling molecules in vivo. NTM infections are often difficult to treat because of the development of resistance. Understanding host–pathogen interactions and defining the role of signalling pathways that can be modulated for host defences for NTM infections are much needed.

## 4. Materials and Methods

### 4.1. Chemicals and Reagents

NS398, BMS-34541, AH6809, and AH 23848 were purchased from Cayman Chemical (Ann Arbor, MI, USA). Antibodies against COX-2 (#12282) and α-tubulin (#2144) were obtained from Cell Signaling Technology (USA). Antibody against actin was obtained from Santa Cruz Biotechnology (Santa Cruz, CA, USA). Antagomir against miR-155, Cy3™ Dye-Labelled Anti-miR™ Negative Control #1 were purchased from Thermo Fisher Scientific (Waltham, MA, USA).

### 4.2. Bacterial Strain

The *Mycobacterium avium*-intracellulare complex strain is a clinical isolate from patients at Emory University Hospital and was kindly provided by Dr. Eileen M Burd. The strain was cultured in Middlebrook 7H9 broth (BD-Difco, Franklin Lakes, NJ, USA), as previously described [63]. Colony-forming units (CFUs) were determined on Middlebrook 7H11agar plates (BD-Difco).

### 4.3. Ethics Statement

The source of human monocytes was the whole blood of healthy donors, and the study (Healthy Donor protocol, IRB# 58507) was approved by the Ethics committee and the Institutional Review Board of Emory University, and informed written consent was obtained from all donors. All human subjects were adults.

### 4.4. Animal

C57BL/6J mice, 6–8 weeks old, were purchased from the Jackson Laboratory (Bar Harbor, ME, USA). All animal procedures were reviewed and approved by the Institutional Animal Care and Use Committee (IACUC) at Atlanta VA Medical Centre, and the approval IACUC number is V021-17. All animal experiments in this study conform to the Institutional Animal Care and Use Committee (IACUC) Guidelines, as well as National Institutes of Health (NIH) and USDA policies on the care and use of animals in research and teaching.

### 4.5. Cell Culture and Treatment

Mouse bone-marrow-derived macrophages (BMDM) from C57BL/6 J were prepared as described previously [64]. Mouse antagomir and corresponding negative control were purchased from Thermo Fisher (USA). Transfection of antagomir was performed using Lipofectamine™ RNAiMAX (Thermo Fisher).

Isolation of human monocytes and culture of macrophages were performed as previously described [65,66]. Briefly, mononuclear cells were isolated by Ficoll-Hypaque Plus (GE Healthcare, Chicago, IL, USA) gradient, cultured in Dulbecco’s Modified Eagle Medium supplemented with 10% heat-inactivated FBS (HyClone Laboratories Inc, Logan, UT, USA), and 50 ng/mL human M-CSF (R&D Systems, Minneapolis, MN, USA). Seven days later, purity of macrophages was assessed by flow cytometry (>90% CD14^+^), as described previously [65].

For in vitro experiments, cells were treated with MAI strain at Multiplicity of Infection (MOI) 25 or 50 for 6 or 24 h. Control cells were treated with sterile phosphate buffered saline (PBS).

### 4.6. MiRNAs Primer-Profiling PCR Array

Total RNA containing miRNAs was extracted through the mirVana miRNA Isolation kit (Thermo Fisher, USA), and miScript II RT Kit (Qiagen) was used for reverse transcription of total RNA containing miRNA. A PCR primer assay was performed using the QuantiTect SYBR Green PCR Master Mix (Qiagen, Hilden, Germany) and gene-specific primers that attached to the bottom of the mouse inflammatory response and autoimmunity-based miScript miRNA PCR arrays (Qiagen) in the ABI 7500 Fast System (Thermo Fisher). PCR primer assay data were analysed on the Qiagen analysis website (www.qiagen.com/us/shop/genes-and-pathways/)and a scatter plot result was the output.

### 4.7. RT^2^ Profiler™ PCR Array

Total RNA was isolated and a cDNA template was synthesized as described previously. Then, the cDNA template was loaded on an RT^2^ Profiler PCR Array (Qiagen), which included 84 genes functionally involved with the cellular oxidative stress response. The samples were run on an ABI 7500 fast system (Applied Biosystems, Carlsbad, CA, USA), and data analysis was performed using the RT^2^ Profiler PCR Array Data Analysis software version 2.5 (Qiagen).

### 4.8. Quantitative RT-PCR

RT-qPCR primers and PCR mix were purchased from Thermo Fisher Scientific (USA). Each sample was analysed in triplicate, using an ABI 7500 FAST PCR System (Thermo Fisher), and the expression values were normalized against the small housekeeping RNAs, U6 snRNA or GAPDH.

### 4.9. Western Blot Analysis

Western blot analysis was performed as previously described [64]. Harvested cells were lysed with RIPA buffer (Millipore Sigma, Burlington, MA, USA) containing 1% Triton X-100 and protease inhibitors.

### 4.10. ELISA

Endogenous Prostaglandin E2 (PGE2) released into supernatants of cell culture medium was assessed by the ELISA kit (Catalog#KGE004B, R&D Systems, USA) according to the manufacturer’s manual.

### 4.11. RNA Binding Protein Immunoprecipitation (RIP) Assay

The Magna RIP RNA-binding protein immunoprecipitation kit (Millipore Sigma) was used for RIP assay. Briefly, after treatment, cells were lysed in 100 µL RIP lysis buffer. The whole cell extract was incubated with anti-Argonaute 2(Ago2) antibody (ab32381, Abcam, Cambridge, UK), or negative control antibody (rabbit IgG, Millipore Sigma) in RIP buffer containing protein A/G magnetic beads. After washing to remove unbounded materials, total RNA was isolated from the precipitate, and qPCR was performed.

### 4.12. Bacterial Survival Assay

Mouse macrophages were plated at a density of 50,000 cells in a 24-well plate. NS398 (10 µM), AH6809 (EP2 antagonist, 10 µM), or AH 23848 (EP4 antagonist, 10 µM) was added to the wells separately, and mixed by gentle pipetting. Cells were then infected with MAI at MOI of 25 for 2 or 24 h. Bacteria outside the cells were removed by gently washing with PBS. Post-infection cells were washed as above and lysed by using 0.1% Triton X-100 in PBS for 5 min at room temperature. Lysates were plated on 7H11 agar and incubated at 37 °C. Bacterial colonies were counted, and viable bacteria in macrophages as CFU/mL were calculated.

### 4.13. Statistical Analysis

Statistical analyses were performed using GraphPad Prism software version 5.0 (GraphPad Software, La Jolla, CA, USA). All experiments were repeated at least three times. Data are expressed as mean ± SEM, unless specified. One-way ANOVA with Turkey’s multiple group comparisons, and two-way ANOVA with Sidak’s multiple comparison test were used, and a *p*-value < 0.05 was considered significant.

## Figures and Tables

**Figure 1 pathogens-10-00920-f001:**
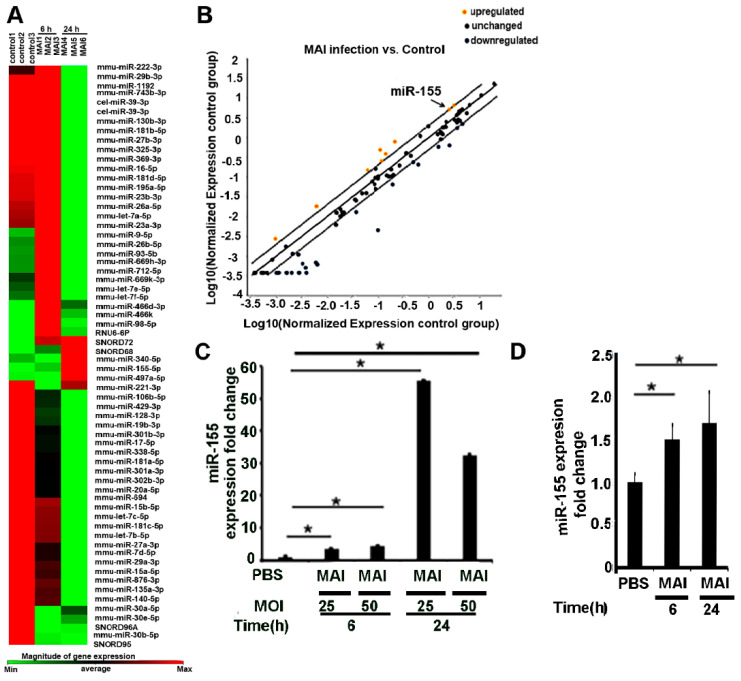
MiR-155 is upregulated in murine and human macrophages by MAI infection. (**A**) Mouse BMDM were infected with MAI (MOI:25) for 6 h. Cells were lysed to assess the profiling of miRNAs using the inflammatory response-mScript miRNA PCR array. A heatmap showed that miRNAs expression levels were altered in BMDM after MAI infection for 6 h. (**B**) Yellow dots indicate increased miRNAs, and blue dots indicate decreased miRNAs (more than twofold). MiR-155 is labelled by the black arrow. (**C**) BMDM were infected with different MOI for indicated time. Cells were lysed and miR-155 expression level was detected via RT-qPCR. (**D**) Human monocyte-derived macrophages were infected with MAI (MOI of 25) for indicated time, then lysed to assess the expression level of miR-155. Using a one-way ANOVA test, all data are presented as mean ± SEM, * *p* < 0.05.

**Figure 2 pathogens-10-00920-f002:**
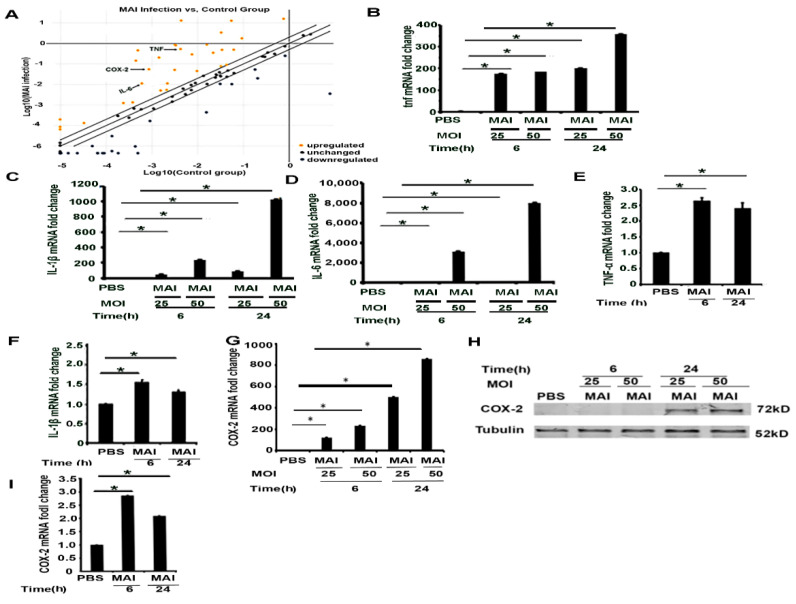
Infection of murine and human macrophages with MAI activates inflammatory cytokines and the COX-2 signaling pathway. Mouse BMDM were infected with MAI (MOI: 25) for 6 h. (**A**) Cells were lysed to assess the profiling of mRNA using the inflammatory responses and autoimunity-RT2 profile PCR array. Yellow dots indicate increased mRNA, and blue dots indicate decreased miRNAs (more than twofold). Data are form *n* = 2 biological replicates. COX-2 is labelled by a black arrow. BMDM were infected with MAI (MOI: 25 or MOI: 50) for 6 h and 24 h. mRNA levels of TNF, IL-1B, and IL-6 were determined by quantitative PCR, as shown in (**B**–**D**). Human monocyte-derived macrophages (MDM) were infected with MAI for 6 h and 24 h. Expression levels of TNF and IL-1β in MDM were detected, (**E**,**F**). BMDM were infected with MAI (MOI: 25 or MOI: 50) for 6 h and 24 h. mRNA and protein levels of COX-2 was determined by quantitative PCR and Western blot, as shown in (**G**,**H**). MDM were infected with MAI (MO1: 25) for 6 and 24 h. The mRNA level of COX-2 was quantified by quantitative RT-qPCR in MDM treated with or without MAI, as shown in (**I**). The one-way ANOVA test was used, and all data are presented as mean ± SEM, * *p* < 0.05.

**Figure 3 pathogens-10-00920-f003:**
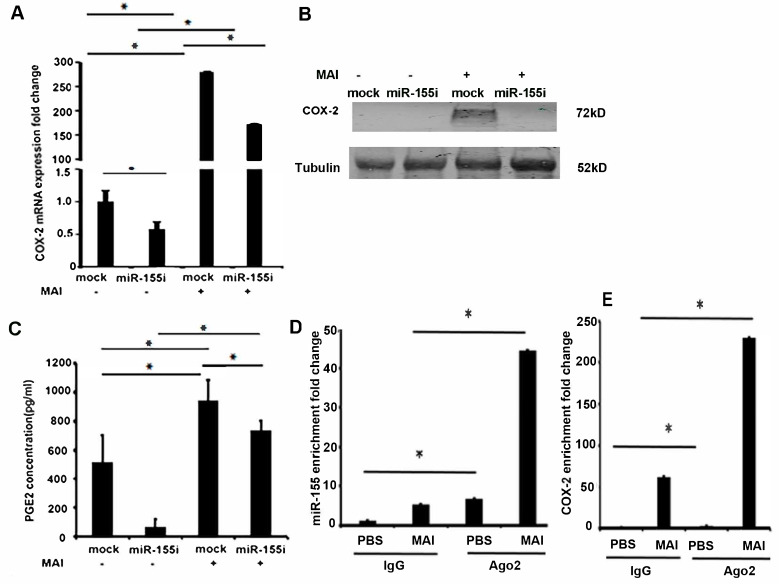
Induction of COX-2 signaling by MAI is dependent on miR-155 in macrophages. BMDM were transfected with antagomir against miR-155 and control. Then, cells were infected with MAI at MOI of 25. (**A**) Expression of COX-2 mRNA was detected through RT-qPCR. (**B**) Cell lysates were immunoblotted with anti-COX-2 antibody, and Tubulin as the internal control. (**C**) Concentration of PGE2 in the supernatant of cells was evaluated through ELISA. Binding of Ago2 with miR-155 (**D**) and COX-2 mRNA (**E**) was detected by RNA immunoprecipitation (RIP) using anti-Ago2 and RT-qPCR. Using a two-way ANOVA with a multiple comparisons test, all data are presented as mean ± SEM, mean ± SEM. * *p* < 0.05.

**Figure 4 pathogens-10-00920-f004:**
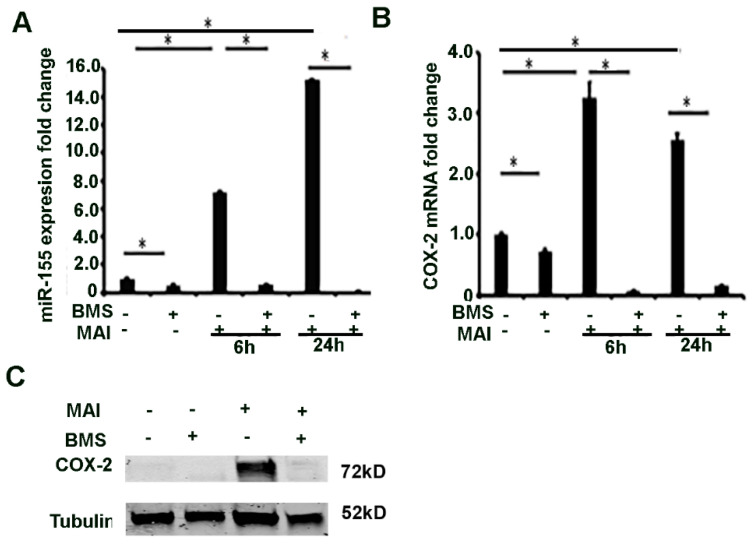
MiR-155 and COX-2 expression are transcriptionally regulated by via NF-κB in macrophages infected with MAI. BMDMs were treated with NF-kB inhibitor (BMS-34541, 10 µM), then infected with MAI (MOI of 25) for 6 h and 24 h. The expression level of miR-155 and COX-2 was evaluated with quantitative RT-qPCR (**A**,**B**). One-way ANOVA with Tukey’s multiple comparisons test was used, and all data are presented as mean ± SEM, * *p* < 0.05. Cell lysates were immunoblotted with anti-COX2 antibody, α-Tubulin as control (**C**).

**Figure 5 pathogens-10-00920-f005:**
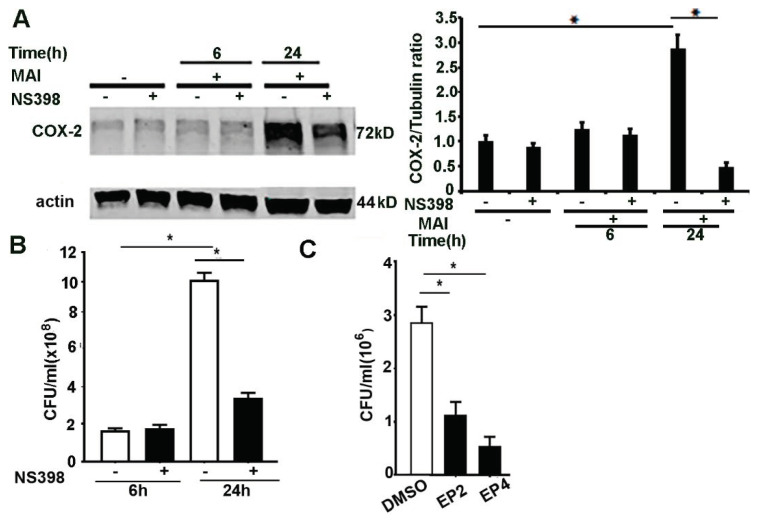
Inhibition of COX-2, EP2, and EP4 enhances killing of ingested MAI in macrophages. BMDM were pre-treated with NS398 (10 µM), then infected with MAI (MOI: 25) for 6 h and 24 h. Cell lysates were immunoblotted with anti-COX-2 antibody, and actin as internal control, as in (**A**). Viable ingested MAI in BMDM was determined at 24 h, as in (**B**). BMDM were pre-treated with the EP2 inhibitor and EP4 inhibitor (10 µM), then infected with MAI (MOI: 25). Viable ingested MAI in BMDM was determined at 24 h, as in (**C**). One-way ANOVA with Tukey’s multiple comparisons test was used, and all data are presented as mean ± SEM, * *p* < 0.05.

**Figure 6 pathogens-10-00920-f006:**
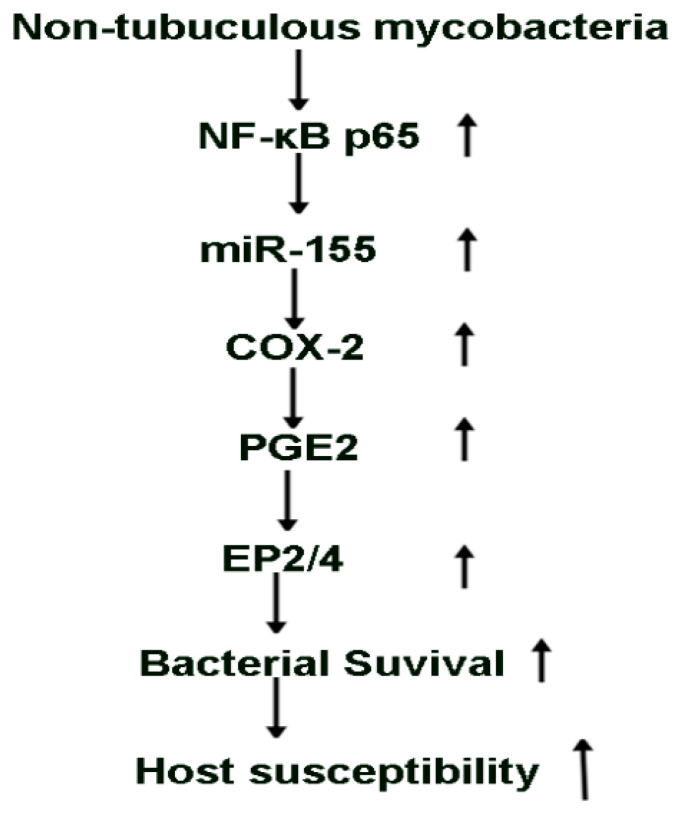
Schematic representation of the proposed model.

## Data Availability

All data generated or analyzed during this study are included in this published article.

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
