# Peer review of "MicroRNA-155 Modulates Macrophages’ Response to Non-Tuberculous Mycobacteria through COX-2/PGE2 Signaling"

_pathogens, 2021, doi:10.3390/pathogens10080920_

Round 1
Reviewer 1 Report
The authors demonstrated that expression of miR-155 and COX-2 increased in macrophages from mice and healthy volunteers that are infected with Non-tuberculous mycobacteria. Additionally, provide novel mechanistic insights into the role of miR-155/COX-2/PGE2/EP signaling and suggest that induction of these pathways enhances the survival of mycobacteria in macrophages. The objectives are clear and the methodology is appropriate to generate the results necessary to determine the conclusion. However, I have some minor suggestions to improve the results and text presentation:
- Line 107 - It is better to show all secondary data in a separate table in the supplementary material than “data not shown”.
- Figure 1B – A volcano plot will be useful to decipher the gene expression regulation.
- Line 141 – Sometimes in the text, the mediator representative name is confused whether it related to the gene or protein nomenclature, such as "tnf/TNF".
- Figure 2 A- Provide a list with fold-change significance of all inflammatory genes analyzed (down- and up-regulation) in the supplementary material.
- Figure 2- Indicate the correct nomenclature of genes and proteins in the graphs axes.
- Line 159- Remove PGE2 from the legend as it was not quantified in this set of experiments.
- Line 201 –“we performed” was written twice.
- Figure 3- I would like to discuss the biological characteristic of the inhibition gene expression of COX-2 with miR-155i, witch remains high, despite a significant decrease compared to mock, but the protein expression is totally prevented. What is the role of MiR-155 in protein translation?
- Line 252 - Apparently, there is a positive looping of COX-2 activity inducing COX-2 expression, so how the expression of mir-155 can be inserted into this loop?
- Figure 6 - Rather than suggesting a flow scheme, it would be more didactic to represent a pathway-to cell scheme in a biological context.
Author Response
Please see the reply below

Reviewer 2 Report
Yuan and collaborators describe one of the mechanism through which macrophages may kill non-tuberculous mycobacteria, using Mycobacterium avium intracellulare as a representative microorganism of this group performing in vitro infection assays with monocyte derived macrophages of mouse an human origin.
The manuscript is well organized and written although many errata have been spotted and should be thoroughly edited for these errors. In the acctached of verson of the manuscript I have highlighted some of the the lines that should be changed.
I suggest homogenizing terms. Sometimes RT-PCR, sometimes quantitative RT-PCR, others qTR-PCR. I suggest using RT-qPCR which is widely accepted.
In Figure 1A heatmap, why are results MAI1, 2 and 3 so different to MAI 4, 5 and 6? An I missing labeling?
How was MAI grown for in vitro assays? How was it quantified in order to adjust for correct MOI for in vitro infection assays?
In Figure 5. Why do you have CFU of 1E8 after 24 hrs in B and then they go down to 1E6 in C at the same time point? Figure 5 legend needs rephrasing.
Line 447. States “at different time points as indicated in the graph”. In which graph? Figure 5?

Author Response
Please find the reply below.

Reviewer 3 Report
In the current manuscript the authors show that MAI infection in murine BMDM results in increase of miR-155, COX-2, and PGE2. Inhibition of miR-155, COX-2 and PGE2 receptors led to decreased intracellular CFU.
- The manuscript needs to be proof-read regarding grammar, typos, sentence part duplications, and interpunctuation.
- Line 91-96 should be integrated in the introduction.
- Line 112: This sentence is unclear and/or can be left out.
- Fig 1A is unclear: What is control 1-3 (group 1), MAI1-3 (group 2), MAI4-6 (group 3). Why are they grouped? Clarification should be provided in figure legend.
- All graphs: Student’s t test is not appropriate since the graphs depict more than two groups and sometimes more than one factor. Appropriate significance tests would be two-way ANOVA with Sidaks multiple comparison test and one-way ANOVA.
- All figure legends should state how many times the experiments were independently repeated and if the graphs show pooled data or one representative experiment.
- All figure legends should state the time post infection of depicted graphs if it is not already depicted in the graph.
- Sentence starting in line 147: “We have previously shown that…” should be moved to introductions and/or discussion.
- Line 187: The sentence starting with “We and others have shown that…” only provides a self-citation for “We have shown…” but not for “and others”.
- Sentence starting in line 192 is incomplete.
- Fig 3E misses the “E“.
- Fig 3E needs re-scaling of the y-axis so differences between group PBS-IgG and PBS-Ago2 are visualized.
- Line 247: This statement is not correct since COX2 and PGE2 signaling has been investigated in M. tuberculosis
- This reviewer is puzzled how the pharmacological inhibition of COX2 by NS398 results in decreased overall protein levels of COX2 since, even though inhibited, the protein should still be there and detectable, at least it is in other publications e.g., Nature PMID: 14760389, Fig 1.
Discussion
- Several statements made in the discussion part need citations.
- That induction of COX2 signaling upon MAI infection id dependent on miR-155 was only shown for mouse BMDM but not human MDM (Fig. 3).
- That expression of miR-155 and COX2 is regulated by NF-kB upon MAI infection was only shown for mouse BMDM but not human MDM (Fig. 4).
- Enhanced mycobacterial killing by macrophages upon COX2 or PGE2 receptor inhibition was only shown for mouse BMDM but not human MDM (Fig. 5).
- For human MDM, only increase of miR-155, COX-2, IL-1b, and TNF-a gene expression upon MAI infection was shown. The abstract and discussion reads as the findings presented in all figures and the proposed model was also checked to be valid for human MDM.
- 6: “Host susceptibility” should also have an upwards arrow.
- 6 can be omitted.
Minor
- I suggest introducing the abbreviations PGE3 and EP2/EP4 in the abstract
- I suggest adding the keyword “Non-tuberculous mycobacteria”.
- Line 50: “Mycobacterium tuberculosis”
- Line 53: “…in Mtb infection…”
Other Suggestions:
- Provide pictures showing complete Westernblots in supplements instead of only cut-out bands?
- Usually microarray raw data must be provided via upload to a platform and made accessible?
Author Response
Please see the comments below.
